# On Demand Biosensors for Early Diagnosis of Cancer and Immune Checkpoints Blockade Therapy Monitoring from Liquid Biopsy

**DOI:** 10.3390/bios11120500

**Published:** 2021-12-07

**Authors:** Sai Mummareddy, Stuti Pradhan, Ashwin Kumar Narasimhan, Arutselvan Natarajan

**Affiliations:** 1Department of Biology and Chemistry, Emory University, Atlanta, GA 30322, USA; sai.sreehith.mummareddy@emory.edu; 2Department of Microbiology, Immunology, and Molecular Genetics, University of California, Los Angeles, CA 90095, USA; stutipradhan@ucla.com; 3Department of Biomedical Engineering, SRM Institute of Science and Technology, Chennai 603203, India; ashwinkn05@gmail.com; 4Molecular Imaging Program at Stanford (MIPS), Department of Radiology, Stanford University, Stanford, CA 94305, USA

**Keywords:** cancer markers, immune checkpoints, PD-1, PD-L1

## Abstract

Recently, considerable interest has emerged in the development of biosensors to detect biomarkers and immune checkpoints to identify and measure cancer through liquid biopsies. The detection of cancer biomarkers from a small volume of blood is relatively fast compared to the gold standard of tissue biopsies. Traditional immuno-histochemistry (IHC) requires tissue samples obtained using invasive procedures and specific expertise as well as sophisticated instruments. Furthermore, the turnaround for IHC assays is usually several days. To overcome these challenges, on-demand biosensor-based assays were developed to provide more immediate prognostic information for clinicians. Novel rapid, highly precise, and sensitive approaches have been under investigation using physical and biochemical methods to sense biomarkers. Additionally, interest in understanding immune checkpoints has facilitated the rapid detection of cancer prognosis from liquid biopsies. Typically, these devices combine various classes of detectors with digital outputs for the measurement of soluble cancer or immune checkpoint (IC) markers from liquid biopsy samples. These sensor devices have two key advantages: (a) a small volume of blood drawn from the patient is sufficient for analysis, and (b) it could aid physicians in quickly selecting and deciding the appropriate therapy regime for the patients (e.g., immune checkpoint blockade (ICB) therapy). In this review, we will provide updates on potential cancer markers, various biosensors in cancer diagnosis, and the corresponding limits of detection, while focusing on biosensor development for IC marker detection.

## 1. Introduction

Cancer is one of the leading causes of death worldwide. The high incidence of cancer and the corresponding elevated mortality rate has made the creation of new diagnostic tools and therapeutic techniques a high clinical priority. Prognosis is best when cancers are detected early, but this task is challenging due to the difficulties of detecting small, early-stage tumors [1]. Tumor-specific signature markers expressed on the cell surface, i.e., cancer biomarkers, can help in diagnosing cancer early. Combining these cancer marker signals with biosensing tools (known as biosensors) should enable early cancer detection [2]. 

As a device, biosensors integrate analytes, receptors, transducers, and outputs to measure the expression level of the cancer markers. These analytical and diagnostic devices can be used to process biological samples (analytes) to collect specific information using a combination of biological detecting molecules (receptors) and an electronic sensor system with a transducer (Figure 1). Biosensors are developed based on the target analyte, such as cancer markers, immune markers, and DNA, found in biological samples. Some examples of biosensors include bio-computers, glucometers, biochips, and resonant mirrors [3].

In this schema we have depicted the significant parts in sequence used in for any biosensor device. It shows analytes from biological samples input to signal output from digital displays by providing a few examples of analyte molecules such as proteins, DNA and other targeting agents; receptors expressed on cells, enzymes, nucleic acid, and antibodies [4]. In addition, this illustration indicating the well-known transducers used in biosensors, i.e., optical, electrochemical, piezoelectric, and thermal. Specific information on each parts of biosensor is provided in the corresponding sections of this review.

Biosensors are designed to detect specific biological markers or analytes (i.e., proteins, DNA, RNA and cells) and convert biological molecule interaction-signals into an electrical signal that can be measured as a digital output. In addition, biosensor technology has the potential to deliver fast and accurate information, as well as measure cancer cells and cancer metastasis. It can also be used to determine the therapeutic effectiveness of anticancer drugs, assess cancer biomarkers, and determine effectiveness of drugs at various target sites. Biosensors are an emergent tool for various disease management with great potential in cancer detection and monitoring. Overall, these biosensors are made to reduce the diagnostic time for a patient’s disease and to monitor therapeutic outcomes. 

There are several reviews available elsewhere with respect to the biosensor device instrumentation, technology, and engineering for signal processing [5,6,7]. Hence, in this review, we would like to illuminate two key areas related to biological components: (a) the various biomarkers used as analytes to measure the disease conditions and (b) types of sensors and transducers that could detect biomarkers at the lowest level. The focus is to further the understanding of ICB and their potential use in the biosensor development. These ICBs play a major role in understanding the early diagnosis and treatment efficacy post chemo or radiotherapy. A detailed analysis of check point markers is discussed for use in future biosensor development.

## 2. Key Methods for the Detection of Cancer Biomarkers from Liquid Biopsy 

Detection of biomarkers and immune checkpoints in liquid biopsies can be analyzed with various transduction principles, targets, and analytes, as summarized in Figure 1. Analytes are designed to bind to a specific receptor, protein or biomarkers on cells. Binding between the target and analyte depends on the transduction principle and the sensor type which produces the corresponding output. Finally, the output signal is processed and analyzed to display on the device. Several techniques are widely used for the detection of various disease markers, such as optical, electrochemical, piezoelectric, and thermal based sensors. Among these detection methods, optical and electrochemical based biosensors are cost effective and highly sensitive with low detection limits and high reproducibility. The basic principle of optical biosensors works on the interaction between antigens and antibodies, where the binding affinity intensity is transformed into proportional electronic signals detected in the transducer unit. The optical sensing unit consists of a laser source, a spectrometer, an immobilized sensor, and an electronic device to amplify the interaction [8]. Optical sensors work based on two methods: (1) Direct detection and (2) Indirect detection through exogenous agents. In both methods the primary signal measurements are derived from changes in the absorption and fluorescence intensity, colorimetric, mechanical sensors such as microcantilevers and variations in refractive index [9]. Furthermore, these detection systems can be applied by combining the opto-electronic device and lab on chip technologies. 

Similarly, electrochemical-based detectors convert chemical energy into electric potentials. To measure this chemical energy, electrodes are used as a transducer, i.e., electrodes are coated with receptors that interact with analytes. When a redox reaction occurs between the analyte and the receptor, the external voltage is applied to the transducer element (electrode), which generates a current. This current is further amplified via signal process tools to identify the desired chemical reaction [10]. Electrochemical sensors are classified based on the detection output method, such as amperometric, potentiometric and voltametric, as well as on enzymatic and non-enzymatic-based detection in liquid biopsies [10,11]. In the next section, we will discuss the various proteins and antigens used for the detection of tumors. 

Detecting cancer using biomarkers from circulating fluids such as the blood has received remarkable attention in recent years. Liquid biopsy sampling is minimally invasive and requires a small volume of blood to detect and characterize tumors and monitor treatment outcome [12]. A broad variety of cancer markers and associated assays have been under development to diagnose cancer via different detecting systems. For example, various blood-based biomarkers such as tumor associated antigens (TAAs, Figure 2), circulating tumour cells (CTCs), and circulating cell-free tumor DNA (ctDNA) are widely utilized. 

Biosensors are designed to detect specific biological markers or analytes (i.e., proteins, DNA, RNA and cells) and convert biological molecule interaction-signals into an electrical signal that can be measured as digital output. In addition, biosensor technology has the potential to deliver fast and accurate information, as well as measure cancer cells and cancer metastasis. It can also be used to determine the therapeutic effectiveness of anticancer drugs, assess cancer biomarkers, and determine effectiveness of drugs at various target sites. Biosensors are an emergent tool for various disease management with great potential in cancer detection and monitoring. Overall, these biosensors are made to reduce the diagnostic time for a patient’s disease and to monitor therapeutic outcomes. There are several reviews available elsewhere with respect to the biosensor device instrumentation, technology, and engineering for signal processing [5,6,7]. Hence, in this review, we would like to illuminate two key areas related to biological components: (a) the various biomarkers used as analytes to measure the disease conditions and (b) the types of sensors and transducers that could detect biomarkers at the lowest level. The focus is to further the understanding of ICB and their potential use in the biosensor development. These ICBs play a major role in understanding the early diagnosis and treatment efficacy post chemo or radiotherapy. A detailed analysis of check point markers is discussed for use in future biosensor development.

### 2.1. Tumor-Associated Antigens (TAAs)

TAAs can be divided into three primary categories: (1) normal proteins that are overexpressed by cancer cells, (2) differentiation antigens, and (3) cancer testis antigens (CTA) (Table 1) [13]. TAAs (Figure 2) and tumor-specific antigens (TSAs) have been studied for decades due to their presence during early stages of cancer. While TSAs are found solely on cancer cells, TAAs are also found on healthy cells but are overexpressed on cancer cells, allowing them to be used to detect cancer. In most instances, the biological system may produce antibodies to combat the TAAs, thereby providing additional opportunities for diagnostic markers. In 1943, Gross et al., conducted an experiment determining the ability of acquired immunity to combat malignant tumors by detecting TAAs [14]. Since then, various TAAs have emerged as potentially effective biomarkers for early detection of cancer. However, rigorous validation protocols are required to make these potential TAAs into early cancer biomarkers.

An ideal TAA should be specific to a type of cancer—it should not be measurable in control or normal tissues, in patients with other tumors, or in those with autoimmune diseases. However, many TAAs lack this specificity. For example, p53 is a tumor suppressor mutated in cancer patients, but it is also detectable in various tumors and in autoimmune diseases [15]. Therefore, rigorous biomarker validation is necessary [16]. Determining the immunogenicity of TAAs allows antigens with the greatest potential for diagnostic and therapeutic purposes to be separated from those with low specificity and sensitivity. When detecting TAAs using biosensing devices, it is important to choose biomarkers with attributes such as high sensitivity, specificity, stability, and reproducibility in the biological system. Furthermore, the signal corresponding to the marker should provide insights into the origin, stage, and progression of disease. For example, Silva and his coworkers have attempted the detection of TAAs such as peanut agglutinin in serum samples at 0.01 mg mL^−1^, which is a comparatively lower limit of detection (LOD) than other sensors [17]. Thus, the determination of TAAs using biosensor devices at low LOD is still in its infant stage. 

### 2.2. Overexpressing Cancer Biomarkers

Several TAA biomarkers are overexpressed in cancer cells and a few examples are described as follows. Mucin-1 (MUC1) is a protein biomarker overexpressed in 90% of breast tumors. The N-terminal of MUC1 is shed from carcinoma cells and can be found in the plasma of women with breast cancer [18]. However, MUC1-N prevalence is greatest in breast cancers that have metastasized, so it is not an ideal target for early detection. On the other hand, human epidermal growth factor receptor-2 (HER2), a well-studied protein overexpressed in cancer cells are potential for targeted therapy [19]. Overexpression of HER2 has been found to directly correlate to poor prognosis of numerous cancers, including breast, esophageal, gastric, ovarian, and endometrial cancer [20]. HER2 overexpression is found in 15–30% of breast cancers which can have up to 25 to 50 copies of the HER2 gene, resulting in nearly 2 million receptors on the tumor cell surface [21]. The extracellular domain of the HER2 protein is cleaved and can be detected in blood serum using ELISA. However, when these antigens have relatively low expression, targeting them during treatment can cause toxicity in healthy tissues. For example, the number of receptor copies expressed on breast cancer cells can be classified as HER2 positive, negative or equivocal, based on the IHC results [21]. 

### 2.3. Differentiation Antigens

Differentiation antigens are expressed by cells during specific stages of cell development and in tissue samples. These differentiation antigens are considered as a new type of cancer biomarker for biosensing. Their specificity indicates a strong potential for differentiation antigens to be detected for accurate cancer diagnosis. For example, glycoprotein-100 (gp100) is a differentiation antigen presented on melanocytic cells in large amounts that can be targeted by anti-melanoma cytotoxic T lymphocytes (CTLs). Additionally, gp100 is immunogenic, and it can induce an immune response, suggesting that melanocyte differentiation antigens may be tumor rejection antigens [22].

Carcinoembryonic antigen (CEA) is another differentiation antigen widely used for cancer screening protocols as an elevated level of serum CEA corresponds to colorectal cancer (CRC) in approximately 17–47% of individuals [23]. It can be used as an independent prognostic factor and can predict outcomes of patients with stage II CRC. CEA has also shown greater sensitivity for detecting recurrence in comparison to other diagnostic techniques such as CT scans. In a study performed by Tsikitis et al., postoperative surveillance of CEA detected first recurrences in 29.1% of patients with early-stage colorectal cancer, while CT scan detected 23.6%, colonoscopy detected 12.7%, and chest X-rays detected 7.3% [24]. The sensitivity of CEA makes it an ideal biomarker to detect colorectal cancer. It is the best-known marker to detect tumor recurrence. Monitoring CEA levels in patients who have recovered from CRC could be an effective screening method (for recurrence). Nevertheless, while CEA has the greatest sensitivity when compared to other diagnostic techniques, it is still unable to detect most CRC recurrences, indicating a need for better tumor-associated antigens to be discovered. Several biosensors have been developed to increase the efficiency of the detection of CEA using nanoparticles. Using anti-CEA as targeted moieties with different sensing mechanism like electrochemical, optical, chemiluminescence, and acoustic wave biosensors were studied [25]. However, there are still restrictions in the detection limit at pico to nano gram level from the samples.

### 2.4. Cancer Testis Antigens

Cancer testis antigens (CTA), classified under tumor germline antigens, are found in the male germ cells of healthy adults. However, CT antigens are also found in tumor cells of various types of cancer, classifying them as neoantigens, or a protein that forms on cancer cells previously unknown to the body. CTA antigens lack MHC class 1 molecules. This allows them to be immune privileged, meaning that they can tolerate the introduction of antigens without causing an immune response. However, the cellular and humoral immune responses observed in cancer patients displaying CTA antigens as well as the correlation of CTA antigens with cytolytic activity of tumor immune infiltrates suggests that CTA antigens are highly specific immune targets [26]. Melanoma-associated antigen-A (MAGE-A) is a CT antigen expressed in 32% of melanomas and 45% of squamous cell carcinomas. The high expression of MAGE-A indicates that it may be a potential target for detection and immunotherapy [27]. However, it was observed that MAGE-A is more prevalent in metastatic cancers than primary tumors, indicating its lack of efficiency towards early detection.

### 2.5. Prostate Specific Antigen (PSA) Based Biosensor 

Prostate specific antigen (PSA) is the most common cancer marker for the detection of prostate cancer from serum. The PSA biosensor is primarily developed with a combination of the PSA targeting receptors and a transducer (e.g., electrochemical, micro-cantilever, and surface plasmon resonance) to detect prostate cancer. 

The changes in vibrational frequency upon the interaction of an antigen binding to an antibody are used to detect PSA [28,29]. Several approaches have been investigated to build biosensors to diagnose prostate cancer non-invasively with greater sensitivity than the standard ELISA technique [30]. 

Integration of conventional biosensors with nanomaterials increase the sensitivity and specificity of such markers. For instance, the use of gold electrodes, graphene or graphene-oxide, carbon nanotubes, hybrid nanoparticles, and other types of nanoparticles are potential agents for PSA detection. The most attractive sensing schemes are summarized in a Table 2. 

Among approaches that results in the highest sensitivity is the electrochemical-based biosensor (as it can detect PSA at attomolar concentrations), followed by mass cantilever sensing and the electro-chemiluminescent (EC) approach. EC-based biosensors can be modified by changing the antibodies and by using other binders such as aptamers, peptides, and nanobodies with the same device [45]. For example, horseradish peroxidases (HRP), alkaline phosphatase (ALP), glucoamylase, *β*-galactosidase, or glucose oxidase are conjugated to antibodies (Ab) and can be employed. In this approach, the Ab_[secondary]_ is directly tagged to the reporter enzymes, which catalyze the reduction of their respective substrates in the presence or absence of a mediator [46]. The HRP enzyme-linked electrochemical sandwich assay is a well-known and widely used assay due to its stability and fast reaction kinetics, which catalyzes the reduction of H_2_O_2_ in the presence of hydroquinone (HQ) [47]. However, it produces a background signal in the absence of the analyte due to endogenous electrochemical activity of its H_2_O_2_ substrate. Additionally, HRP relies on the use of mediators for signal generation. ALP has been used to overcome the background and mediator limitations of HRP.

The EC-based indirect detection of PSA assay was developed with incorporation of multiple receptor elements, NPs, redox labels, or other biomolecular processes. These PSA biosensors provided a basis for multi-target analysis with increased sensitivity and selectivity. For example, the detection limit demonstrated clinically relevant LODs ranging from 1 ng mL^−1^ to 0.020 fg mL^−1^ (Table 2). Using these devices, a linear range of 0.001–10 ng mL^−1^ and a limit-of-detection (LOD) of 0.84 pg mL^−1^ were obtained for detecting PSA in serum. In related study, Chen et al. developed gold screen-printed electrode-based microfluidic devices (GSPE-MFDs) to demonstrate the detection and quantification of PSA in human serum samples [48]. In this system, GSPE Ab_c_ conjugated magnetic particles (MNPs) were immobilized for capturing PSA, followed by HRP-Ab_s_ captured on the PSA-attached electrode using the microfluidic device. The enzymatic and electrochemical reaction between HRP and HQ generates current signals for measurement. Similarly, Zani et al. utilized a magnetic bead-based assay by using mouse IgG-Ab_c_ for PSA capture and ALP tagged anti-mouse IgG for signal transduction [49]. This PSA-targeted biosensor was able to detect as low as 2 ng mL^−1^ in human serum samples. Furthermore, it shows a linear range of PSA detection (1–80 ng mL^−1^). Overall, enzyme-based sandwich assays demonstrated very high specificity to PSA, but they also have intrinsic enzymatic instability issues [50]. In addition, these assays require several washing steps to remove non-specific binding to avoid false negative or false positive results.

## 3. Immune Checkpoints as a Marker for Cancer Diagnosis

Immune checkpoints and their use in cancer immunotherapies is a rapidly growing field of study [51]. Immune checkpoint-based treatment increases the hope among the cancer patients and especially those who did not respond to more established treatments [51]. Currently, the surface markers, programmed death-1 (PD-1) and CTLA-4 are well known for their function in the immune system, as well as their role in cancer as theranostic applications [52]. PD-1 and CTLA-4 are two major checkpoints approved by the FDA for immune checkpoint blockade therapy [53]. Similarly, other checkpoints, including TIM-3, BTLA, and LAG-3, are under active investigation as potential biomarkers for cancer theranostics [54]. These immune checkpoints can be detected on tumor cells and in the tumor microenvironment (TME) through traditional tissue biopsies and IHC as well as via biosensors. Although these biomarkers are potentially for the detection of cancer, the developmental status of biosensors designed to detect these immune checkpoint markers is still at an early stage [55]. In this article, we will focus upon several such soluble immune checkpoints-sPD-1, sPD-L1, sLAG-3, and sTIM-3-that are expressed on peripheral blood mononuclear cells that could potentially be diagnosed through biosensors.

### 3.1. T-Cell Immunoglobulin and Mucin Domain-3 (TIM-3)

TIM-3 is part of the TIM family and is a receptor found on interferon-γ-producing CD4+ and CD8+ T cells [56]. Initially, TIM-3 was considered primarily as a co-inhibitory checkpoint; a protein that is co-regulated and co-expressed alongside PD-1, LAG-3, and TIGIT [57]. Later, TIM-3 emerged as a co-stimulatory checkpoint depending on the cell type on which it is expressed and the immune response. It is also involved in T-cell exhaustion and therefore found in tumors [58]. In a study of leukocytes from peripheral blood mononuclear cells, the level of TIM-3 expression was significantly higher in the CD4+ and CD8+ T cells of ovarian cancer patients. CD4+ T-cells displayed further elevation of TIM-3 in ovarian cancer recurrence, suggesting TIM-3 as a biomarker for early detection as well as detection of recurrence [59].

TIM-3 upregulation in CD4+ and CD8+ peripheral T cells was found to correlate with the presence of osteosarcoma. The level of TIM-3 expression also increased with tumor stage and metastasis, and higher TIM-3 levels were associated with worse overall survival. Such findings suggest TIM-3′s value as a diagnostic and prognostic marker for osteosarcoma [60]. In hepatocellular carcinoma, TIM-3 expression is induced by other cytokines located in proximity in the tumor microenvironment (TME). However, TIM-3 expression, along with the expression of other T cell exhaustion markers such as PD-1 and CTLA-4, is lower in the non-tumor microenvironment (NTME) and the peripheral blood than the TME [61]. The variation in expression of TIM-3 in the TME and NTME presents a challenge for using immune checkpoints to detect cancer non-invasively. 

### 3.2. B- and T-Lymphocyte Attenuator (BTLA)

BTLA is a lymphocyte inhibitory receptor found on B-, T-, and other mature lymphocyte cells. BTLA regulates the immune system by inhibiting T cell reactions and restricting cytokine production and creates a tumor microenvironment that suppresses immune responses [62]. Reports indicating high BTLA expression were found in gastric cancer, pancreatic cancer, and chronic lymphocytic leukemia (CLL)/small lymphocytic leukemia [63]. However, this upregulation does not correlate to the presence of BTLA in the bloodstream. For example, a higher gene transcript level is observed in CLL patients. BTLAs were detected in low level peripheral blood B cells and normal BTLA levels were detected on T cells. Even in-vitro stimulation resulted in lower levels of BTLA expression on B and T lymphocytes [64].

Different stages of B and T cells developed from bone marrow, which include pro-B/T, pre-B/T and immature B/T cells, each express unique marker. T cells are activated by B cells and further divided into T-helper and T-regulatory cells. In 1994, the number of B-cells were determined using piezoelectric sensors by immobilizing anti-B cells on the surface electrode [65]. However, capturing B-cells at high accuracy requires sensitive biosensors to obtain an amplified signal. T-cell lymphocytes CD4+ detection using nano based biosensors was explored recently by Carinelli et al. using magnetic particles [66]. Here, isolation of CD4+ cells were captured magnetically via anti-CD4+ using magnetic particles [66]. Similarly, single wall carbon nanotubes-based detection coupled with electrochemical biosensor for the detection of CD4+ using with a limit of detection at 1 × 10^2^ cells mL^−1^ [67]. Chen and his coworkers developed a rapid assay biosensor to detect dynamic T-cell activation using nanoparticles, wherein, polyaniline fibers coated with gold nanoparticles detected multiple immunosensing markers such as CD69, CD25, and CD71 with specific anti-CD molecules. Time dependent activation of immune markers were captured with an LOD of 1 × 10^4^ cells mL^−1^ [68]. Hence, understanding the role of T-cell immuno-sensors at high LOD is important to reveal the immune inhibition of activated drugs.

### 3.3. Immune Markers in Serum

Immune markers serve as potential prognostic biomarkers in serum and could have significant predictive power of disease progression in personalized medicine. While some immune checkpoints can be detected on peripheral blood cells (e.g., PD-1) and others (PD-1, PD-L1, LAG-3, and TIM-3) can be detected in serum due to ectodomain cleavage. This section reviews such potential immune markers for the serum.

### 3.4. Lymphocyte-Activation Gene 3 (LAG-3)

A disintegrin and metalloproteinase domain-containing protein (ADAM) such as ADAM-10 and ADAM-17 cleave the extracellular domain of LAG-3 [69]. Higher levels of serum LAG-3 (sLAG-3) were found to correlate with earlier stages of non-small cell lung cancer. Serum of patients with stage III and IV cancer had lower levels of sLAG-3 comparatively [70]. Contrastingly, when serum sLAG-3 levels were assessed in gastric cancer patients, individuals with gastric cancer were found to have sLAG-3 levels of 247.52 ± 51.28 ng/mL while healthy individuals had sLAG-3 levels of 869.46 ± 64.35 ng/mL. Additionally, sLAG-3 was found to have a greater sensitivity and accuracy in comparison to other serum biomarkers for gastric cancer such as CEA. sLAG-3 had a sensitivity of 88.60% and an accuracy of 90.65% when evaluated at a threshold value of 378.330 ng/mL. On the other hand, CEA had a sensitivity of 56.20% and an accuracy of 68.61% at a threshold value of 21.755 ng/mL [71]. Measures of soluble LAG-3 in serum have prognostic value, for example, high levels of soluble LAG-3 in some subsets of breast cancer correlate with disease-free, metastasis-free, and overall survival [72].

### 3.5. PD-1

Serum PD-1 or sPD-1 (along with sPD-L1) is elevated in the serum of melanoma patients in comparison to healthy blood donors [73]. Higher sPD-1 levels are also found in non-small cell lung carcinoma (NSCLC), diffuse large B-cell lymphoma (DLBCL), chronic lymphocytic leukemia (CLL), nasopharyngeal carcinoma (NPC), HCC, pancreatic adenocarcinoma, and advanced rectal cancer [74]. Additionally, sPD-1 and sPD-L1 can be used to predict the outcome of certain immunotherapies, such as anti-PD-1 treatment. A threshold of 500 pg/mL was used to differentiate progression-free survival rates, because of the finding that high baseline serum sPD-1 levels were associated with poor survival [73]. Similarly, serum sPD-1 and sPD-L1 levels were significantly elevated in triple-negative breast cancer patients prior to neoadjuvant chemotherapy. However, patients who responded well to the treatment (as observed through complete or partial remission) had decreased serum sPD-1 and sPD-L1 levels compared to those who had a poor response to the treatment [75].

### 3.6. PD-L1

The soluble programmed-death ligand 1 (aka PD-L1) is an immune biomarker that can potentially be used for early cancer diagnosis research due to its important role in cancer immunoregulation. The PD-L1 marker essentially binds to its receptor of PD-1 and partakes in pleiotrophic signaling pathways. When PD-L1 binds to the PD-1, there is a halt in the T cells going out to kill the foreign antigens. Thus, studies have shown that the inhibition of this interaction between the PD-L1 marker and the receptor PD-1 can result in drastic reduction of tumor growth and promote antitumor immunity. Thus, it is important to detect the concentrations of the PD-L1 marker in blood. However, it is difficult to quantify the concentration of PD-L1 marker in blood due to signal sensitivity at low concentrations.

Localized surface plasmon resonance (LSPR) biosensors enable one to improve the sensitivity of the signal at low concentrations of PD-L1 using metal nanoparticles [31]. The detection of PD-L1 can be seen through the wavelength shift in the biosensor where the anti-sPD-L1 antibodies get immersed and localized with the gold nanoparticles in the biosensor. In this study, gold nano-shells were developed with LSPR at NIR region around 1600 nm to detect PD-L1. At different serum media, the biosensor showed high specificity to PD-L1 with a sensitivity of 1 pg mL^−1^. Thus, gold nano shells-based detection of PD-L1 show high sensitivity compared to other conventional techniques like ELISA, IHC and PCR. PD-L1 undergoes proteolytic cleavage by ADAM 10 and ADAM 17 in breast cancer, resulting in a ~37 kDa fragment that can be detected in the media [76,77]. sPD-L1 is elevated in patients with papillary thyroid cancer (PTC) compared to healthy patients. Serum sPD-L1 levels were found to be 0.37 ng/mL in healthy individuals and 0.48 ng/mL in cancer patients. Furthermore, higher sPD-L1 levels were indicative of shorter disease-free survival, suggesting the value of sPD-L1 as a prognostic marker of PTC [78]. Even in primary central nervous system lymphoma, serum sPD-L1 levels (0.429 ng/mL) were higher than those in healthy individuals (0.364 ng/mL). Higher sPD-L1 levels resulted in more frequent relapse (78% of the study participants) than the lower sPD-L1 levels (50%) [79].

ADAM 10 and ADAM 17 are also major target of TIM-3 that result in soluble TIM-3 (sTIM-3) [80]. Serum sTIM-3 is found to be higher in patients with osteosarcoma when compared to patients with benign tumors and the control group (14.4 ng/mL ± 2.9 vs. 10.3 ng/mL ± 1.7 vs. 6.3 ng/mL ± 1.9). Benign tumors also displayed higher sTIM-3 levels than the control, suggesting a possibility for early detection for prevention [81]. Additionally, elevated sTIM-3 levels were also associated with hepatocellular carcinoma (HCC). The control group had a mean serum sTIM-3 level of 2.64 ± 0.32 pg/mL while patients with HCC had a mean serum sTIM-3 level of 3.57 pg/mL ± 0.22 [82].

## 4. Biosensors for Cancer Markers and Immune Checkpoint Detection

Biosensors that have been used in the detection of cancer and immune checkpoint markers include electrochemical and colorimetric biosensors. Most of these biosensors share the following characteristics: consumer friendly, easy to use systems, moderate cost, low turnaround time for the results, and minimal sample size required for testing. However, when compared to detecting markers in samples with low concentrations these biosensors are not as sensitive or specific in comparison to the SPR biosensors [83,84]. Furthermore, these types of sensors can only be used once or a limited number of times, reducing their value for conducting multiple runs.

PD-1 and PD-L1 are known [85,86] to be independent prognostic factors for several tumor associated markers for immunotherapy. Kruger et al. quantified sPD-1 and sPD-L1 in pancreatic cancer serum with ELISA using the human PD-1 antibody duo sets for ELISA development [87]. Reporting on immune check point of therapy with PD-L1 alone is insufficient and showed poor outcome in pancreatic cancer. For example, higher concentrations of PD-L1 and PD-1 were identified in 41 patients to test for advanced stages of carcinoma. Further studies are required to understand the correlation between the two prognostic factors for immune check point blockade therapy. Similarly, a multiplex immunoassay detection (8-color flow cytometry) method was developed using cryopreserved samples of healthy human donors to determine the level of CD8+ and CD4+, for early cancer diagnosis [88]. Upon stimulation of T cell lymphocytes, measurement of upregulated soluble isoforms of LAG-3, TIM-3, CTLA-4, and PD-1, as detected in flow cytometry, directly correspond to treatment response.

### 4.1. Biosensors for Detection of Multiple Cancer/IC Markers

Multiplexed immune checkpoint biosensor (MICB) development is progressing rapidly due to their high specificity and selectivity [3]. Multiplex biosensors require small volumes of sample to diagnose the disease and work simultaneously to detect multiple immune checkpoints. The parallel detection device is growing as it can be potentially translatable for routine clinical applications. However, with these sensors, the analyte tends to take longer to bind with the cell surface without the use of nanofluidic mixing. Hence, these sensors are time consuming and generate a response slowly. For example, a droplet loaded in the electrode can detect the PD-L1 marker and can run multiple samples at once (28 for the multiplexed biosensor on the PD-L1 marker) [42]. Advanced versions of multiplexed electrohydrodynamic biosensors have shown improved performance for biomarker detection [89,90]. However, many of the multiplexed lateral flow nanosensors display this limitation. Additionally, getting and visualizing the target alone for these sensors is also difficult as there are many non-specific interferences displayed in the system. Based on the applications and sample results, biosensors can be broadly classified into two types: (a) high sensitivity and specificity at the cost of a complex system (SPR and multiplexed biosensors) and (b) easy to use and cost and time effective, but not possessing high sensitivity and specificity (colorimetric and EC biosensors).

Another MICB developed by Wuethrich et al. detects soluble PD-1, PD-L1, and LAG-3 in parallel in liquid biopsies, using as little as a single sample drop of approximately 20 µL volume per target immune checkpoint [42]. MICB utilizes a microfluidic sandwich immunoassay with high-affinity yeast cell-derived single chain variable fragments (ScFvs) as well as alternating current electrohydrodynamic in-situ nanofluidic mixing. The high-affinity yeast cell-derived ScFvs are an alternative to the use of monoclonal antibodies and polyclonal antibodies, which are costly and have a limited shelf life. High-affinity yeast cell-derived ScFvs provide longer shelf life without sacrificing specificity and can be mass-produced for a lower cost. The nanofluidic mixing stimulates immune checkpoint interactions with the biosensor while diminishing non-target sensor binding, providing greater specificity for immune checkpoint capture.

MICB has ability to analyze 28 samples in less than two hours. To understand the ability for MICB to detect soluble PD-1, PD-L1, and LAG-3 in liquid biopsies, various concentrations of immune checkpoints were spiked into diluted human serum. While the assay was able to detect a differentiable signal response from the samples with the immune checkpoints, fortified samples had a concentration equal to the prognostic cut-off, which may differ from the threshold for cancer detection [42].

### 4.2. Exosome Biosensor

Exosomes are the key players in communication at the intercellular level through surface biomarkers. For instance, exosomal proteins are fundamental in cancer development, drug resistance, and differentiation etc [91]. Expression of PD-L1 in exosomes is a biomarker that is overly expressed in lung tumors. Hence, many exosomal proteins that include exosomal immune markers like PD-L1, can be promising cancer biomarkers. In a recent study, researchers constructed a compact surface plasmon resonance biosensor that detects exosomal PD-L1 and EGFR [92]. Looking at the different serum samples of patients with lung cancer, researchers used this biosensor to determine the specificity and sensitivity of the biomarker in these samples.

This biosensor worked through optical resolution using laser beams. Essentially, two laser beams were used in which one allowed for the SPR to interact with the exosomal proteins on the surface of the sensor. The other beam was used as reference, and the two photodetectors quantified the exosomal proteins by magnifying the intensity of the initial beam and the reflection beam. Overall, in terms of results, the SPR biosensor was much more effective in detection of exosomal PD-L1 in comparison to conventional methods like ELISA [93]. First and foremost, the study depicts that the SPR biosensor was very accurate and effective in detecting the PD-L1 in serum samples including that of a Stage III lung cancer patient; however, the method of ELISA was unable to detect PD-L1 in general. These serum samples were all 50 microliters. Additionally, through these biosensors, researchers proposed that the exosomal PD-L1 can be a biomarker (although this still needs to be confirmed through larger sample sets) for lung cancer diagnosis as the SPR biosensor detected much more exosomal PD-L1 in lung cancer patients compared to the normal controls [92]. The sensitivity and specificity of the SPR biosensor was much greater than the ELISA testing as the ELISA test could not detect any of the PD-L1 in serum samples in which there was a very low abundance of the exosomal PD-L1.

### 4.3. Biosensors for Overall Cancer Immunotherapy

Biosensors could play a pivotal role in augmenting cancer immunotherapy for treatment monitoring and evaluation of disease condition. For example, several study results revealed that the nano biosensor can be utilized cytokine secretion from individual cells for hours [94]. A specific biosensor is combined with a gold nanohole array sensor and a microfluidic-based system for accurate depiction and detection of cytokine secretion within a cell-to-cell basis. Additionally, some groups have worked on detection of the IL-2, IFN-γ, and TNF-α cytokines using gold nanorod-based biosensors [95]. Using these different types of sensors, researchers showed that immunoanalysis could be done on a smaller volume of analyte and a shorter timeframe than conventional ELISA methods [96]. Thus, these nanomaterial-based biosensors can be employed for more versatile applications than detection of a major immune checkpoint marker such as PD-L1.

### 4.4. Biosensors for CRISPR

Clustered regularly interspaced short palindromic repeats (CRISPR/Cas) based biosensors have also become more important for rapid detection of nucleic acids, exosomes, tumor DNAs, etc. CRISPR/Cas sensors use Cas effectors to cleave a specific DNA sequence. One of the main effectors, Cas9, has become well known due to its effective ability of editing genomes, identifying nucleic acids, and regulating transcription [97,98]. Specifically, Cas9 uses single guide RNA to help target a specific double-stranded DNA and cleave it for the purpose of selective genome editing [97,98]. In addition, the usage of nuclease-deactivated Cas9 can allow for binding and rebinding to the targeted double-stranded DNA, which is essential as fluorescent enzymes can be fused to the DNA to allow for detection of specific nucleic acid sequences due to the fluorescence and light emission [83,84]. This fusion of integrating Cas9 with a fluorescent enzyme was instrumental for researchers to detect DNA sequences of the American Zika virus. Other types of Cas effectors such as Cas12 and Cas13 have also been explored. Wherein, Cas12 effector is extremely important due to its ability to detect single-stranded DNA, which adds more to the detection of the different nucleic acids rather than focusing only double-stranded DNA with Cas9 effector. However, the main reason researchers have started to focus on CRISPR/Cas-based biosensors for cancer detection is due to the Cas13a effector. This effector works with the collateral cleavage activity of single-stranded RNAs. When there is a specific RNA of interest, the Cas13a effector cleaves RNA, the collateral cleavage of the reporter RNA (reRNA) which allows for the detection of the RNA with accurate quantification [85]. This effector is key for recognition of many miRNAs which can be either up-regulated or down-regulated. Hence, by being able to detect levels of miRNAs rapidly and effectively, this field of CRISPR/Cas-based biosensing is becoming a hot topic and displays promise for the early detection of cancer. However, one key limitation is that these biosensors involve sophisticated systems to run a tedious process to function fully [97,98]. Hence, reading these biosensors and their detection of the specific analyte requires well-trained professionals to understand the process and readout. Furthermore, the sensitivity of the biosensors is relatively low when compared to the other amplification methods [97,98]. Though many researchers are working on CRISPR/Cas-based biosensors, none of them could meet the needs of in vivo testing due to low sensitivity detection. Despite these issues, there is potential for the development of biosensor based on the Cas13a effector and the collateral cleavage activity detection [99].

### 4.5. Biosensors for ICB: Advantages and Limitations

Globally, the market size and demand for biosensors is rapidly increasing [98]. In 2026, the market size is expected to expand to about 37 billion dollars for all the different types of biosensors (electrochemical, FRET, piezoelectric, optical, etc.). However, improvements in biosensors are still a challenge because of the low concentration of biomarkers expressed over the cells. Nanomaterials-based detectors have the potential to amplify the signals with limit of detection at pico- or femtomolar concentration. In addition, the COVID-19 pandemic in 2020 has shown us the potential need of biosensors. Hence, in this review we have outlined current improvements and strategies in biosensors over the last decade for biomedical applications have been explored.

The detection of immune checkpoints via liquid biopsy presents an attractive option for early cancer detection. Currently, immune checkpoints are not widely used for diagnostics. Instead, they are utilized as prognostic markers for cancer and as predictors for the success of various immunotherapies. The biosensors proposed in recent years from the studies of Xu et al. [25] showcase the various advantages of using these biosensors for detecting small concentrations of PD-L1. For the quantification of different immune checkpoint markers like PD-L1, the studies showcased the limitations of ELISA and immunohistochemistry. For example, with ELISA, it is difficult to detect PD-L1 with very small concentrations. However, PD-L1 in multiple types of serum media is found in very small concentrations and thus, ELISA becomes extremely inaccurate in many situations. Immunohistochemistry is also an employed method that becomes very time-consuming and has very complex procedures. With these surface plasmon resonance-based biosensors and the multiplex biosensor, the studies show the simple-to-use nature of the biosensors, the very high specificity and sensitivity of the sensors, as well as their time and cost effectiveness. One way to ensure cost-effectiveness is a sensor that uses very cheap single-chain variable fragments from yeast cell lines in replacements of the very costly antibodies of the different immune checkpoint proteins that are used in the conventional methods.

Biosensors have been developed to detect the concentrations of the immune checkpoint markers in serum samples e.g., PD-L1 [42], but other potential IC markers can be identified through same approach. These biosensors demonstrated a few unique features with fine differences in the benefits and limitations of using one over the other. Table 2 and Table 3 list the various advantages and disadvantages of these biosensors that are reported in the relevant literature [31,42,83,84,90]. Developments in surface plasmon resonance (SPR)-based biosensors development, especially for the detection of immune checkpoint markers in early detection of cancer, have progressed well due to several key advantages [31,92].

LSPR biosensors show very high specificity and sensitivity (LOD: 1–5 pg/mL) for the immune checkpoint marker PD-L1 [31]. Furthermore, these biosensors are highly selective for the marker in comparison to the normal detection methods, e.g., immunohistochemistry, and can be integrated with a label-free detection system. Most of these sensors use fluorescent labeling or some other type of labeling approach to analyze the different analytes. However, the detection of analytes was required with different types of media which resulted in non-specific binding and high background signals. In addition, receptor labeling requires extensive time, effort, and optimization for the development of accurate biosensors for reproducible results. These limitations do limit the biosensors’ usability. However, there is no additional labeling step required since this biosensor relies on physical properties of the analyte immobilized on a surface. Hence, these biosensors do not require nearly as much labor nor time to work effectively. Additionally, the biosensors allow for real time detection of the analyte as the analyte is localized into a compact surface; thus, there is no alteration of binding sites which could possibly occur with biosensors that require a labeling system.

SPR-based devices can cause configurational changes of the marker or analyte being detected by biosensors [32]. These biosensors use a mechanism of action where the analyte gets immobilized on the surface, often causing lower binding affinities due to the configurational changes of the immobilized receptor which, can cause the analyte binding ability and make it hard to detect the analyte accurately. SPR biosensors detect the marker or analyte by a change in wavelength with the aid of gold nanomaterials. Using these nanomaterials and immobilizing the analyte is a complex system to use for the commercialization of the product on a large scale.

## 5. Conclusions and Future Potential of Biosensors for ICB Detection

In this review, we have provided a detailed account on potential cancer markers and emerging IC markers, as well as their role in the tumor microenvironment (TME) in relation to cancer diagnosis. We presented the current state of IC markers’ potential, in association with biosensors for the diagnosis and monitoring of various diseases and outlined the opportunities and applications of biosensors for improvements in healthcare. Thus, measuring and monitoring these IC markers in very minimal concentrations could assist in assessing early cancer. We have tabulated a list of biosensors applications in multiple studies and their limit of detections.

However, even though these biosensors have been proposed and built, it is important to note that only PD-L1 and CTLA-4 immune markers are currently approved for the treatment of ICB. Consequently, the future holds enormous potential for biosensor development to quantify other immune markers in cancerous microenvironments. Furthermore, surface plasmon resonance-based biosensors alongside nanoparticle usage are emerging as the most promising types of biosensors in quantification of these immune checkpoint markers. From other studies, the drawback of using these types of sensors is that they may be extremely complex to understand. Thus, a key objective, going forward, is to explore how to make these biosensors into user-friendly devices for cancer theranostics by using ICB.

In summary, in this review we clearly highlight the ongoing development of biosensors and their future potential for disease management, particularly for the early detection of cancer. In the future, these complex biosensors should be refined, as they should be able to measure with high sensitivity and simultaneous measurement of multiple biomarkers. In addition, it is imperative that the engineering and fabrication of biosensors should be easily accessible, low cost, and less complex to operate as a routine clinical testing tool for the detection of specific biomarkers. Furthermore, nanomaterial and nanoparticle-based biosensors have become prominent for early detection of cancer due to their high sensitivity as stated earlier in this review. However, additional focused investigations are required for the stability of these nanoparticles for prolonged periods with consistent and accurate readout.

## Figures and Tables

**Figure 1 biosensors-11-00500-f001:**
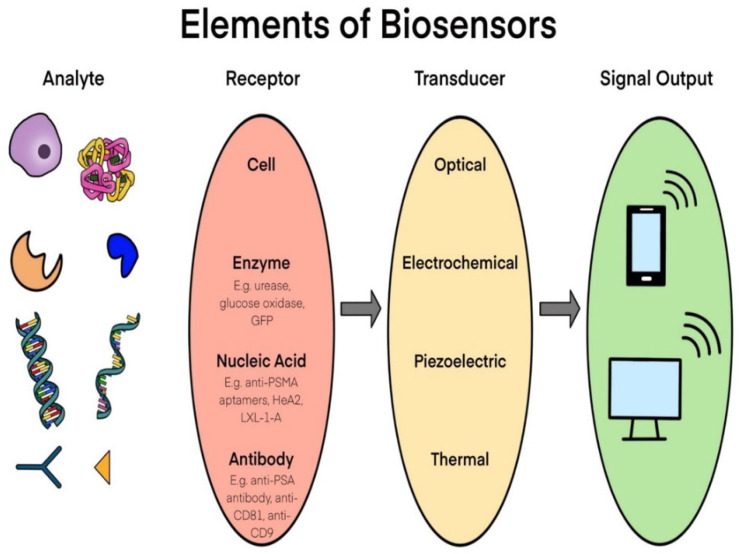
General schema of biosensor device workflow.

**Figure 2 biosensors-11-00500-f002:**
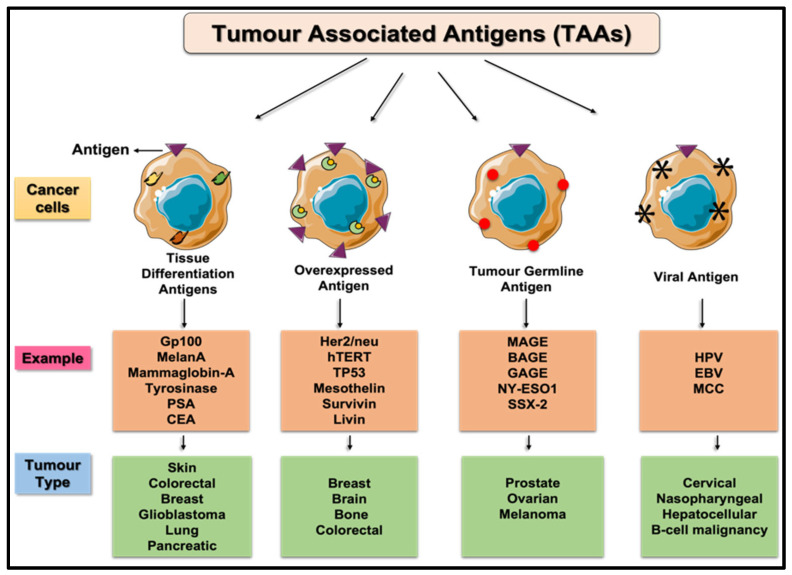
Tumor associated antigens and its expression pattern.

**Table 1 biosensors-11-00500-t001:** Candidates for Early Detection Biomarkers to Cancer Phenotype: Advantages and Limitations.

**Types of Antigens**	**Antigen**	**Cancer Type**	**Strengths for Early Detection**	**Limitations for Early Detection**
**Overexpressed Antigen**	MUC1	Breast	Expressed in over 90% of breast tumors	Most prevalent in metastasized breast cancers
	HER2	Breast/Esophageal Gastric/Ovarian Endometrial	Nearly 2 million receptors expressed on tumor cell surface	Can cause toxicity to healthy cells given low expression of antigen
**Differentiation Antigen**	gp100	Melanoma	High expression in malignant glioma cells	Expression in normal brain tissues
	CEA	Colorectal cancer	Greater sensitivity than other diagnostic methods	Limited sensitivity Most effective in detection of recurrent cancers
**CT Antigen**	MAGE-A	Melanoma Squamous cell carcinomas	Very specific to cancer cells	Higher prevalence in metastatic cancers

**Types of Antigens**

**Table 2 biosensors-11-00500-t002:** List of Different Biosensors, their Advantages and Limitations and their Potential Biomarker Targets.

Biosensor Types	Advantages	Disadvantages	Examples of Biomarkers Detected	References
Localized and Compact Surface Plasmon Resonance Biosensor (LSPR and CSPR)	Highly sensitive and specific Label-free system Real-time measurement and detection Usage of gold nanomaterials Small sample size required to run	Complex mechanism of action Cost-effectiveness Non-target binding Immobilization on surface causing configurational changes	PD-L1 MT1-MMP IFN-γ PSA IgG TNF-α CRP	[31,32,33,34]
Electrochemical Biosensor	Cost and time- effective Small sample size required to run Very easy to use system for consumers Really good detection limits	Reproducibility issues (either one or few time usage) Not as sensitive as the other biosensors and conventional methods Very low shelf life	CEA NSE MUC1 EpCAM Multiple Types of miRNAs BRCA1 HER2	[35,36,37,38]
Colorimetric Biosensor	Cost and time-effective Very easy-to use Small sample size required Portable and easy to maintain	Low sensitivity and specificity Reproducibility issues Low shelf life Limited multiplexing and quantification capabilities	CEA AFP PSA miRNA-148a miR-21 miR-155	[39,40,41]
Multiplexed Nanobiosensors	Cost-effective High sensitivity and specificity compared to conventional methods Parallel detection of checkpoint markers (gives more insight in checkpoint interactions) Very small sample size needed	Still need to find effective ways for the target to get to the surface rapidly Non-target binding on the surface Non-specific interferences	PD-1 PD-L1 LAG-3 miR-21 miR-574-3p EpCAM Bladder Cancer MicroRNAs	[42,43,44]

**Table 3 biosensors-11-00500-t003:** Biomarkers and associated limits of detections (LOD) when using biosensors.

Cancer Type	Biomarker	Diagnostic (D) Prognostic (P)	LOD/mL	References {LOD References}
**Tumor-Associated Antigens**				
Breast	HER-2 ECD, CEA CA15-3	D/P	2 ng, 5 ng, 21.8U	[99,100,101,102] {[103,104,105]}
Ovarian	CA125, CA15-3	D	35 U, 12U	[21,106,107,108,109], {[106,108]}
Pancreatic	CEA	P	5 ng	[110] {[111]}
	CA19-9	D	37U	[112] {[113]}
	CA125	D/P	35U	[114,115] {[116]}
Gastric	CEA, HER-2 ECD	P	5 ng, 24.75 ng	[110,117] {[117,118]}
NSCLC	CEA	P	3–5ng	[119,120] {[120]}
Endometrial	CA 125	D/P	17.8U	[121,122] {[121]}
Colorectal	CEA	D/P	5ng	[123] {[124]}
Prostate	PSA	D	2.5–4 ng ^#^	[125] {[126]}[127]
**Immune Checkpoint Markers**				
Primary central nervous system lymphoma (PCNSL)	PD-L1	D/P	0.43 ng	[78] {[79]}
NSCLC, DLBCL, CLL, NPC, HCC, ADR, PAC, HCC, Melanoma	PD-1	P	ND, 500 pg ^1^	[73,74] {[73]}
Breast, Gastric	LAG-3	P	120 pg, 378.3 ng	[70,71] {[71,73]}
PAC, Osteosarcoma, Ovarian	TIM-3	P, D/P, ND	3 ng, 14.4 ng	[59,60,81,82] {[81,82]}

**Note:** ND: No optimal cutoff has been established; ^#^ Age dependent, non-small cell lung carcinoma (NSCLC), Diffuse large B-cell lymphoma (DLBCL), Chronic lymphocytic leukemia (CLL), Nasopharyngeal carcinoma (NPC), HCC, Advanced rectal cancer, Pancreatic adenocarcinoma (PAC), References in {} are corresponding to LOD values, ^1^ Melanoma.

## Data Availability

Not applicable.

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
