# Peer review of "On Demand Biosensors for Early Diagnosis of Cancer and Immune Checkpoints Blockade Therapy Monitoring from Liquid Biopsy"

_biosensors, 2021, doi:10.3390/bios11120500_

Round 1

Reviewer 1 Report

The manuscript by Mummareddy et al present OnDemand biosensors for early diagnosis of cancer and immune checkpoints blockade therapy monitoring from liquid biopsy.

The figures and the tables are very nice organized, but the structure  of the manuscript is a little disorganized. The chapters and subchapters are not correctly numbered, and I recommend the subchapter Lymphocyte-activation gene 3 (LAG-3) to be moved before the subchapter PD-1, as ADAM (A disintegrin and metalloproteinase domain-containing protein) definition is presented later on.

Some citations are missing:

“There are several reviews available elsewhere with respect to the biosensor device instrumentation, technology, and engineering for signal processing.”-please provide the citations to the reviews you are mentioning

PD-1 and PD-L1 are known (?) to be independent prognostic factors for several tumor 372 associated markers for immunotherapy”- add reference instead of the “?”

Although presented in Figure 1, not all the receptors given as example are described in the manuscript (e.g urease, GFP). Please add the corresponding references in the text.

The English is good overall, but chapter “6. Biosensors for detection of multiple cancer/IC markers” requires corrections. Please also rephrase: ”Higher levels of PD-L1 and PD-1 is observed in 41 patients to test advanced stages of pancreatic cancer”

Author Response

The manuscript by Mummareddy et al present OnDemand biosensors for early diagnosis of cancer and immune checkpoints blockade therapy monitoring from liquid biopsy.

The figures and the tables are very nice organized, but the structure of the manuscript is a little disorganized.

We acknowledge the reviewer’s comment on table and figure organization, and the structure of the revised manuscript is reorganized as per the reviewer’s suggestions. In addition, we have provided content outline in the revised manuscript.

The chapters and subchapters are not correctly numbered, and I recommend the subchapter Lymphocyte-activation gene 3 (LAG-3) to be moved before the subchapter PD-1, as ADAM (A disintegrin and metalloproteinase domain-containing protein) definition is presented later on.

As per the reviewer’s advice we have moved the LAG-3 subsection prior to the PD-1subsection. Please find the line # s 367-378, in the revised manuscript.

Some citations are missing:

“There are several reviews available elsewhere with respect to the biosensor device instrumentation, technology, and engineering for signal processing.”-please provide the citations to the reviews you are mentioning

We have included additional citations (references # 4-6) for this statement (line #105-106) in the revised manuscript.

Cited References:

[5]        M. Soler and L. M. Lechuga, "Principles, technologies, and applications of plasmonic biosensors," Journal of Applied Physics, vol. 129, p. 111102, 2021.

[6]        Y. T. Chen, Y. C. Lee, Y. H. Lai, J. C. Lim, N. T. Huang, C. T. Lin, et al., "Review of Integrated Optical Biosensors for Point-Of-Care Applications," Biosensors (Basel), vol. 10, Dec 18 2020.

[7]        D. Liu, J. Wang, L. Wu, Y. Huang, Y. Zhang, M. Zhu, et al., "Trends in miniaturized biosensors for point-of-care testing," TrAC Trends in Analytical Chemistry, vol. 122, p. 115701, 2020.

PD-1 and PD-L1 are known (?) to be independent prognostic factors for several associated markers for immunotherapy”- add reference instead of the “?”

New references were added or the following statement “PD-1 and PD-L1 are known (70, 71)…..” at line # 431.

Although presented in Figure 1, not all the receptors given as example are described in the manuscript (e.g Urease, GFP). Please add the corresponding references in the text.

References are included in the revised manuscript.

[4]        Mehrotra P. “Biosensors and their applications - A review”. J Oral Biol Craniofac Res. Vol. 6, pp. 153-159, May 2016.

The English is good overall, but chapter “6. Biosensors for detection of multiple cancer/IC markers” requires corrections.

In the revised manuscript we have updated this subsection at “4.1. Biosensors for detection of multiple cancer/IC markers” line # 442-455.

Please also rephrase:Higher levels of PD-L1 and PD-1 is observed in 41 patients to test advanced stages of pancreatic cancer”

We have rephrased this sentence (line # 434-435) as follows “For example, higher concentrations of PD-L1 and PD-1 were identified in 41 patients to test for advanced stages of carcinoma.”

Reviewer 2 Report

This work proposes an extensive review on biosensors for early diagnosis of cancer and immune checkpoints blockade therapy monitoring from liquid biopsy. Although the manuscript is elaborated from three aspects, including biomarkers in early diagnosis, different biosensors and IC markers detection, it is deficient in logicality and coherence. It is suggested to resubmit after major revision.

  1. The author should readjust the unclear sequence number.
  2. There is a lack of explanation of key method used in the study. The author should better enumerate the various methods and match with corresponding pictures.
  3. The author should explain the outline separately to make the article logical and coherent.

Author Response

  1. The author should readjust the unclear sequence number.

As per reviewer’s suggestion we have fixed the sequence number issues. In addition, we have provided content outline in the revised manuscript (see line # 41-63).

  1. There is a lack of explanation of key method used in the study. The author should better enumerate the various methods and match with corresponding pictures.

We have updated additional paragraphs in the revised manuscript to enumerate the various methods and match with corresponding pictures (Please see the line #s 113-135). For reviewer’s convenience the same content we have reprinted below.

2. Key methods for the detection of cancer biomarkers from liquid biopsy

Detection of biomarkers and immune checkpoints in liquid biopsy can be analyzed with various transduction principles, targets, and analytes, as is summarized in Figure 1. Analytes are designed to bind to a specific receptor, protein, or biomarkers on cells. Binding between the target and analyte depends on the transduction principle and the sensor type which produces the corresponding output. Finally, the output signal is processed and analyzed to display on the device. Several techniques are widely used for the detection of various disease markers, such as optical, electrochemical, piezoelectric, and thermal based sensors. Among these detection methods, optical and electrochemical based biosensors are cost effective and highly sensitive with low detection limits and high reproducibility. The basic principle of optical biosensors works on the interaction between antigens and antibodies, where the binding affinity intensity is transformed into proportional electronic signals detected in the transducer unit. The optical sensing unit consists of a laser source, a spectrometer, an immobilized sensor, and an electronic device to amplify the interaction [7]. The optical sensor works on two methods. 1. Direct detection and 2. Indirect detection through exogenous agents. In both methods the primary signal measurements were derived from changes in the absorption and fluorescence intensity, colorimetric, mechanical sensors such as micro cantilevers and variations in refractive index [8]. Furthermore, these detection systems can be applied by combining the opto-electronic device and lab on chip technologies.

Similarly, electrochemical based detectors convert chemical energy into electric potentials. To measure this chemical energy, electrodes are used as a transducer, i.e., electrodes are coated with receptors to interact with analytes. When a redox reaction occurs between the analyte and the receptor, the external voltage is applied to the transducer element (electrode), which generates current. This current is further amplified via signal process tools to identify the desired chemical reaction [9]. Electrochemical sensors are classified based on the detection output method, such as amperometric, potentiometric and voltametric, as well as on enzymatic and non-enzymatic based detection in liquid biopsy [10]. In the next section, we will discuss the various proteins and antigens used for the detection of tumors.”

  1. The author should explain the outline separately to make the article logical and coherent.

In the revised the manuscript we have provided outline separately prior to the introduction section as a content list. Please see the content outline in the revised manuscript (line # 41-63).

Reviewer 3 Report

OnDemand biosensors for early diagnosis of cancer and immune checkpoints blockade therapy monitoring from liquid biopsy by Sai Mummareddy et al is a well documented review article with provided data and references. However  before the final publication I would recommend to add more discussions on the improvement and future aspects of biosensor development for cancer. Also, it would better to add one subpoint on CRISPR based diagnostics since they are more popular nowadays for cancer diagnostics.

Author Response

However, before the final publication I would recommend adding more discussions on the improvement and future aspects of biosensor development for cancer.

As per reviewer’s suggestion additional discussions were added in the revised manuscript (please see the line #s 603-610).

“In summary in this review, we clearly highlight the ongoing development of biosensors and their future potential for disease management, particularly for the early detection of cancer. In the future, these complex biosensors should be refined, as they should be able to measure with high sensitivity and simultaneous measurement of multiple biomarkers. In addition, it is imperative that the engineering and fabrication of biosensors should be easily accessible, low cost, and less complex to operate as a routine clinical testing tool for the detection of specific biomarkers. Furthermore, nanomaterial and nanoparticle-based biosensors have become prominent for early detection of cancer due to their high sensitivity as stated earlier in this review. However, additional focused investigations are required for the stability of these nanoparticles for prolonged periods with consistent and accurate readout.”

Also, it would better to add one subpoint on CRISPR based diagnostics since they are more popular nowadays for cancer diagnostics.

We appreciate the reviewer’s suggestions and we have added additional sub-section on CRISPR based diagnostics in the revised manuscript (please see the line #s 504-528).

4.4. Biosensors for CRISPR

Clustered regularly interspaced short palindromic repeats (CRISPR/Cas) based biosensors have also become more important for rapid detection of nucleic acids, exosomes, tumor DNAs, etc. CRISPR/Cas sensors use Cas effectors to cleave a specific DNA sequence. One of the main effectors, Cas9, has become well known due to its effective ability of editing genomes, identifying nucleic acids, and regulating transcription [82, 83]. Specifically, Cas9 uses single guide RNA to help target a specific double-stranded DNA and cleave it for the purpose of selective genome editing [82, 83]. In addition, the usage of nuclease-deactivated Cas9 can allow for binding and rebinding to the targeted double-stranded DNA, which is essential as fluorescent enzymes can be fused to the DNA to allow for detection of specific nucleic acid sequences due to the fluorescence and light emission [82, 83]. This fusion of integrating Cas9 with a fluorescent enzyme was instrumental for researchers to detect DNA sequences of the American Zika virus. Other types of Cas effectors such as Cas12 and Cas13 have also been explored. Wherein, Cas12 effector is extremely important due to its ability to detect single-stranded DNA, which adds more to the detection of the different nucleic acids rather than focusing only double-stranded DNA with Cas9 effector. However, the main reason researchers have started to focus on CRISPR/Cas based biosensors for cancer detection is due to the Cas13a effector. This effector works with the collateral cleavage activity of single-stranded RNAs. When there is a specific RNA of interest, the Cas13a effector cleaves RNA, the collateral cleavage of the reporter RNA (reRNA) which allows for the detection of the RNA with accurate quantification [84]. This effector is key for recognition of many miRNAs which can be either up-regulated or down-regulated. Hence, by being able to detect levels of miRNAs rapidly and effectively, this field of CRISPR/cas based biosensing is becoming a hot topic and displays promise for the early detection of cancer. However, one key limitation is that these biosensors involve sophisticated systems to run a tedious process to function fully [82, 83]. Hence, reading these biosensors and their detection of the specific analyte requires well-trained professionals to understand the process and readout. Furthermore, the sensitivity of the biosensors is relatively low when compared to the other amplification methods [82, 83]. Though many researchers are working on CRISPR/Cas-based biosensors, none of them could meet the needs of in vivo testing due to low sensitivity detection. Despite these issues, there is potential for the development of biosensor based on the Cas13a effector and the collateral cleavage activity detection [84].

Round 2

Reviewer 2 Report

The authors have modified the manuscript as suggested. It is suggested to be accepted as this form.